# Adsorptive Removal of Naproxen from Water Using Polyhedral Oligomeric Silesquioxane (POSS) Covalent Organic Frameworks (COFs)

**DOI:** 10.3390/nano12142491

**Published:** 2022-07-20

**Authors:** Suleiman Bala, Che Azurahanim Che Abdullah, Mohamed Ibrahim Mohamed Tahir, Mohd Basyaruddin Abdul Rahman

**Affiliations:** 1Department of Chemistry, Faculty of Science, Universiti Putra Malaysia (UPM), Serdang 43400, Selangor Darul Ehsan, Malaysia; bsulaiman1983@gmail.com (S.B.); ibra@upm.edu.my (M.I.M.T.); 2Department of Physics, Faculty of Science, Universiti Putra Malaysia (UPM), Serdang 43400, Selangor Darul Ehsan, Malaysia; azurahanim@upm.edu.my; 3Integrated Chemical Biophysics Research, Faculty of Science, Universiti Putra Malaysia (UPM), Serdang 43400, Selangor Darul Ehsan, Malaysia

**Keywords:** covalent organic frameworks, naproxen, octa(phenyl)silesquioxane, anti-inflammatory drugs, adsorption

## Abstract

Covalent organic frameworks are porous crystalline compounds made up of organic material bonded together by strong reversible covalent bonds (these are novel types of materials which have the processability of extended or repeated structures with high performance, like those of thermosets and thermoplastics that produce high surface coverage). These have a long-term effect on an arrangement’s geometry and permeability. These compounds are entirely made up of light elements like H, B, C, N, O and Si. Pharmaceuticals and personal care products (PPCPs) have emerged as a new threatened species. A hazardous substance known as an “emerging toxin,” such as naproxen, is one that has been established or is generated in sufficient amounts in an environment, creating permanent damage to organisms. COF-S7, OAPS and 2-methylanthraquionone(2-MeAQ), and COF-S12, OAPS and terephthalaldehyde (TPA) were effectively synthesized by condensation (solvothermal) via a Schiff base reaction (R_1_R_2_C=NR′), with a molar ratio of 1:8 for OAPS to linker (L1 and L2), at a temperature of 125 °C and 100 °C for COF-S7 and COF-S12, respectively. The compounds obtained were assessed using several spectroscopy techniques, which revealed azomethine C=N bonds, aromatic carbon environments via solid ^13^C and ^29^Si NMR, the morphological structure and porosity, and the thermostability of these materials. The remedied effluent was investigated, and a substantial execution was noted in the removal ability of the naproxen over synthesized materials, such as 70% and 86% at a contact time of 210 min and 270 min, respectively, at a constant dose of 0.05 g and pH 7. The maximum adsorption abilities of the substances were found to be 35 mg/g and 42 mg/g. The pH result implies that there is stable exclusion with a rise in pH to 9. At pH 9, the drop significance was attained for COF-S7 with the exception of COF-S12, which was detected at pH 11, due to the negative Foster charge, consequent to the repulsion among the synthesized COFs and naproxen solution. From the isotherms acquired (Langmuir and Freundlich), the substances displayed a higher value (close to 1) of correlation coefficient (R^2^), which showed that the substances fit into the Freundlich isotherm (heterogenous process), and the value of heterogeneity process (n) achieved (less than 1) specifies that the adsorption is a chemical process. Analysis of the as-prepared composites revealed remarkable reusability in the elimination of naproxen by adsorption. Due to its convenience of synthesis, significant adsorption effectiveness, and remarkable reusability, the as-synthesized COFs are expected to be able to be used as potential adsorbents for eliminating AIDs from water.

## 1. Introduction

Covalent organic frameworks (COFs) facilitate the exact incorporation of organic structural components into predesignable hollow scaffolds through topological design [1,2,3,4]. COFs with distinct roles, including semiconduction [5], emission [6], catalysis [7], proton conduction [8] and energy conversion and storage [9], have been produced as a result of advances in topology diagrams and synthetic processes during the previous few years [10,11,12,13,14,15]. Pharmaceuticals and personal care products (PPCPs) have sparked interest around the entire globe due to their widespread utilization [16,17] in regular living and potentially extended half-lives [18]. Multiple PPCPs, for instance, have lately been discovered in groundwater, contaminated water, potable water [19,20], and even waterbodies containing vegetation and fish [21,22]. As a result, PPCPs are regarded as a new category of water contaminant [23,24]. Anti-inflammatory drugs (AIDs) are amongst the most commonly prescribed medications [25,26]. Due to their analgesic, anti-inflammatory, and antipyretic properties, these medications are used to cure illnesses in living organisms [27]. Owing to their strong hydrophilicities and water persistence, AIDs can remain in the aqueous solution for a long time [26]. Furthermore, AIDs are highly resistant to biotic deterioration [28], making them the medicines most commonly found in aquatic environments.

PPCPs like naproxen (NAP) and ketoprofen (KTP) have been removed from water using a variety of methods [29,30], include photodegradation [25], photo transformation [31], and advanced oxidation processes [32], and ozonation [33]. Because of its operational simplicity and economic feasibility, as well as the lack of secondary pollutants created, adsorption may be the most promising technology for removing medicines from water [34]. Numerous materials were used to remove AIDs such as NAP from water, comprising soil or clay [35], carbonaceous compounds like activated carbon (AC) and bone char [36,37,38], nanoporous substances [39,40], metal organic frameworks (MOFs) [41,42], and covalent organic frameworks [43,44,45,46]. COFs have displayed considerable promise in the field of environmental clean-up [47,48,49,50] The homogeneous mesoporous architectures of COFs enable the rapid penetration of tiny adsorbates into their core as adsorbent materials. Furthermore, graphene-based materials (GnO/MOFs) have been successfully used to remediate pharmaceutically polluted wastewater [47]. COF substances have also been developed to tidy up environmental pollutants. However, neither synthetic or pharmacological adsorption of water has been reported thus far in POSS/COF compounds. As a result, more research is required to examine the synergic activity of POSS COFs (COF-S12) regarding adsorption for POSS/COF compounds to be used in pharmaceutical adsorption.

In this study, POSS COF-S7 and COF-S12 materials were prepared and used for the adsorption of AIDs (NAP) from water in attempt to comprehend the adsorption behavior and evaluate their efficacy as prospective adsorbents for the removal of AIDs. The nanomaterials demonstrated exceptional adsorption properties toward the adsorbates (NAP). The mechanisms of adsorption, pH influence, adsorption kinetics, and isotherms are all explored. The overall architecture of the antibiotic naproxen is depicted in Figure 1.

## 2. Materials and Methods

All chemicals and reagents were purchased commercially from Sigma-Aldrich (St. Louis, MO, USA) and used without further purification: octa(phenyl)silesquioxane (OPS), benzene-1,4-dicarboxaldehyde (terephthalaldehyde), fuming nitric acid, tetrahydrofuran (THF), hydrazine hydrate, hexane, ethylacetate, mesitylene, 1,4-dioxane, methanol, ethanol, acetic acid, iron (III) chloride hexahydrate, magnesium sulphate (anhydrous), charcoal powder Pd/C, and celite. The synthesis of octa(nitrophenyl)silesquioxane (ONPS) and, with a small modification, octa(amino phenyl)silesquioxane (OAPS), was carried out, and ONPS was synthesized accordingly [51,52,53,54].

### 2.1. Synthesis of Octa(nitrophenyl)silesquioxane (ONPS)

A 5 g (4.84 mmol) quantity of OPS was applied to a small (30 mL) portion of fuming nitric acid with stirring at 0 °C; once the implementation was completed, the mixture was then stirred for approximately 30 min and then for 20 h at ambient temperature. The mixture was poured onto 250 g of ice bag and filtered via glass wool. It accumulated a slightly yellow precipitate, was treated with water (around 100 mL, five times until the pH was approximately 6) and then washed with ethanol (100 mL, three times). The powder collected was dried in an oven at about 80 °C to eliminate the remaining moisture. This yielded about 4 g (80%) of powder.

### 2.2. Synthesis of Octa(amino phenyl)silesquioxane (OAPS)

Octa (amino phenyl) silesquioxane (OAPS) was synthesized with a slight modification of the method reported in the literature [51,54]. In a 250 mL 3 necked round-bottom container fitted with a mechanical stirrer and a condenser, 3 g of ONPS (2.60 mmol), 120 mg of FeCl_3_·6H_2_O, and 2 g of active charcoal powder were charged. The flask was then supplemented with distilled THF (40 mL). The solution was stirred and heated to 60 °C in a N_2_ atmosphere. Hydrazine hydrate (13 mL) was added dropwise into the solution. The reaction proceeded for 5 h, and then the mixture was ventilated and purified through celite. The filtrate was merged with 25 mL of ethyl acetate and rinsed three times with H_2_O. The organic layer was dehydrated over MgSO_4_ and poured into 250 mL of hexane. The precipitate produced was accumulated by filtration. The substance was redissolved in a mixture of 15 mL THF and 25 mL ethyl acetate and reprecipitated into 250 mL of hexane. The collected powder was vacuum-dried for 24 h. This yielded about 40% off-white powder.

### 2.3. Synthesis of COF-S7

COF-S7 was synthesized by a solvothermal process with the combination of octa (amino phenyl) silesquioxane and 2-MeAQ. In a 7 mL vial, OAPS (0.10 mmol, 115 mg) was charged with 2 mL *N*,*N*′-dimethylacetamide (DMAc). In different vials, the organic linker 2-methylanthraquionone (C_15_H_10_O_2_), (0.15 mmol, 266 mg) was charged with 3 mL DMAc. Both solutions were sonicated for 45 min to obtain a homogeneous solution, then 3–5 drops of 6 M acetic acid was supplemented and placed in a screw-capped glass vial in an oven at 120 °C for 3 days, under N_2_. The material was washed with 10 mL ethanol five times and dried for 24 h with a vacuum desiccator to produce a brown needle-like crystal powder which was dried for 24 h. This yielded about 40% off-white powder.

### 2.4. Synthesis of COF-S12

The synthesis of POSS COF-S12 was carried out by solvothermal reaction of octa (amino phenyl) silesquioxane (OAPS) and terephthalaldehyde (TPA). Typically, in a 7 mL vial, OAPS (0.08 mmol, 92 mg) was charged with 2 mL of mesitylene. In different vials, the organic linker (TPA) (C_8_H_6_ O_2_), (0.15 mmol, 80 mg) was charged with 3 mL of mesitylene and 2 mL of 1,4-dioxane. Both solutions were sonicated for 30 min, then 2–3 drops of 6 M acetic acid were added and the mixture placed in a capped glass vial in an oven at 100 °C for 3 days, under N_2_. The substance was cleansed with 10 mL hexane/ethanol five times and dried for 24 h under a vacuum desiccator to produce a fine reddish-brown powder.

### 2.5. Characterization of the COF Materials

The PXRD information was obtained using a PW 3040/60 MPD X’Pert Pro PAN analytical XRD machine from The Netherlands that used CuKα radiation. The screening was obtained using about 3 to 40 o at 2θ and the wavelength used was weighted with CuKα radiation = 1.5418. These X-rays were generated by a cathode ray, screened to provide a monochromator, collimated to concentrate them, and focused towards the object. All the models, including cell parameters and atomic sites, were produced using the Materials Studio software package, using the Materials Visualizer module. The linker units were primarily located with their centroid at the vertex spots obtained from the Reticular Chemistry Structure Resource (RCSR). The RCSR database was founded in order to postulate a universal system for the nomenclature, classification, identification, and retrieval of topological assemblies and now functions as a system for the identification of crystal nets of interest. This database has been instrumental in terms of both designing new crystalline resources as well as identifying previously documented ones. Accordingly, all the models were constructed in the tetragonal system, with the layers lying on the ab plane. The space groups with the most possible symmetry were preferred. An active minimization was accomplished to improve the geometry of the building units, utilizing the universal force field employed in the *Forcite* module of *Materials Studio*. During this procedure, the unit cell parameters for each model were also optimized. The COF-S7 and COF-S12 models’ respective powder patterns were generated and matched to the experimental designs to establish the best agreement. They were used to refine the unit-cell parameters by accomplishment full-profile-pattern (Pawley) refinements against the experimental powder designs [50].

In the FT-IR spectrum, the spectrophotometer emits infrared beams at the sample and measures the wavelengths absorbed by the sample. Infrared light could pass through the material since it was sufficiently small. The objects can be assessed by comparing spectra from the infrared spectral database. Scanning electron microscopy is used to analyze surface morphology, topographical composition, and crystallographic data. A scanning electron microscope (SEM) examines a material with a beam of electrons to produce a magnified image of the object [55,56,57,58]. The SEM study-sample technique demands that the materials are small enough to fit in the tester stage as the ambient conditions and strong electron beam energy should be confined. Using conductive adhesive, the materials are then mounted tightly onto a sample holder. The analysis was achieved with different magnifications using a JEOL JSM-6400 scanning electron microscope (JEOL, Tokyo, Japan). The TGA analysis in this study was conducted using a Mettler Toledo TGA/STDA851 thermal analyzer (Columbus, OH, USA). Prior to analysis, the sample was deposited in an aluminum pan that was punctured. The material was heated to temperatures ranging from 50 to 800 °C, at a rate of 10 °C/min, and at a pressure of less than 200 cm/nitrogen gas flow. For the NMR spectroscopy, the substance was placed in a magnetic field, and the NMR signal was produced by the excitation of nucleus specimens with electromagnetic radiation, which is monitored by responsive radio frequencies. The intramolecular magnetic force around an atom in a molecule modifies the resonance frequency, allowing access to the electronic framework and distinct functional units of the structure.

### 2.6. Adsorption Experiments

Naproxen (50 mg L^−1^) sequential attenuations were prepared by dissolution in a 90:10 *v*/*v* water-methanol solvent. By offering an essential volume of 0.1 M NaOH aq. solution, the pH level was adapted to 7.0 (taking into account the typical pH of river and rain water [42]. By sequential dilution, NAP solutions at the essential concentrations (5–50 mgL^−1^) was formed from the stock solution. The absorption spectrum of the solution was observed using a UV spectrometer (UV-1650, Shimadzu, Kyoto, Japan) at 272 nm to obtain the concentrations of NAP. Prior to the adsorption, the adsorbent was dehydrated at 100 °C for 12 h in a vacuum oven to disperse water. The adsorbent (5.0 mg) was poured to the adsorbate mixture (25 mL, pH 7.0), and the solution was stirred at 25 °C for 1–5 h in magnetic stirrer at a consistent rate (250 rpm). After the adsorption investigations, the mixture was filtered through a polytetrafluoroethylene syringe filter (hydrophobic, 1 mL). UV spectrophotometry examination was used to quantify the remnant proportion of adsorbates in the solution. The influence of the solution’s pH on NAP adsorption on COFs were further investigated by applying volumes of 0.1 M HCl (or NaOH) solution to the solution. The reproducibility was tested by applying 20 mg of the utilized COFs to ethanol (25 mL), agitating them for 24 h, then isolating them to restore the adsorbents [45]. To totally eradicate the adsorbed NAP, the regenerating procedure was executed five times. Afterwards, the regenerated substance was filtered and dried for 24 h at 100 °C.

## 3. Results and Discussion

We designed two new COFs with a 2-methylanthraquionone and terephthalaldehyde as organic units for COF-S7 and COF-S12, respectively. Both COFs were synthesized by a solvothermal process (Figure 2).

In order to confirm that the phases observed from PXRD measurements for COF-S7 and COF-S12, simulated powder patterns were calculated from these models and compared with the experimentally observed data (Figure 1).

Figure 1 shows the PXRD spectra of as-synthesized COF-S7 and COF-S12 materials, which exhibit notable strong peaks at 2θ degrees and less than 10 (<10). COF-S7 had notable peaks at 8.1, 10.0, 10.6, 16.1 and 20.9°, which corresponded to hump-ordered (112), (200), (202), (3–12) and (107) reflections, respectively. Interestingly, COF-S12 plane reflections were also observed to be hump-ordered to (002), (112), (211), (220), (213), (321), (400), (224) and (215), which correspond to 2θ angles of 5.5, 7.8, 8.8, 10.9, 11.5, 12.7, 13.6, 15.1 and 16.1°.

COF-S7 possesses brown needle-like crystal in the tetragonal *I-4* space group with unit cell parameters a = 17.632, b = 117.632, and c = 33.692, 90°, and volume unit cell 10,475.1 Å^3^. Similarly, COF-S12 is a fine pale-yellowish crystalline powder with a tetragonal space group *I-4*/*mmm*, unit cell parameters a = 27.001, b = 27.001, and c = 31.140, and γ = 90°, and a unit cell volume of 22,703 Å^3^. The scopes from the intensity of the calculated and as-synthesized COFs could be attributed to the non-full connectivity among the central compound (OAPS) and the linkers; therefore, the occurrence of the materials from the incomplete reaction might be present in the crystalline materials acquired.

Chemical stability of the synthesized COF-S7 and COF-S12 materials were explored in various solvents, including methanol, ethanol, acetone, hexane, acetonitrile, chloroform, dichloromethane and tetrahydrofuran (Figure 2). This was done to remove the synthesized solvents used (DMAc and mesitylene/1,4-dioxane) in an attempt to obtain an optimal solvent for the activation.

Furthermore, the structural reliability of the materials was investigated by FTIR, CP MAS NMR solid ^13^C and ^29^Si techniques. The FTIR spectrum for the ONPS showed that the two sharp peaks at 1344 and 1537 cm^−1^ were attributable, respectively, to symmetry and asymmetry υN=O. However, after reduction with hydrazine hydrate, which acted as a reducing agent, as indicated in the FTIR spectrum of OAPS, both strong absorption peaks collapsed, proving that the reduction reaction was achieved. In the OAPS spectrum, new broad peaks were observed at 3352 and 3224 cm^−1^, which were ascribed to primary imine νN–H stretching (Figure 3). A similar study was conducted in which the symmetry and asymmetry υN=O were detected at 1350 cm-1 and 1529 cm^−1^, respectively. Furthermore, νN–H bond stretching were also revealed at 3369 cm^−1^ and 3220 cm^−1^, respectively [51]. Equally, in the ONPS compound, a Si–O–Si bond was observed at 1092 cm^−1^ and a slight decrease was also identified at 1074 cm^−1^ in the OAPS FTIR spectrum. This research was in agreement with that by [51,54], who established that the ONPS and OAPS Si–O–Si bonds occurred at 1119 cm^−1^ and 1100 cm^−1^, respectively.

COF-S7 and COF-S12 exhibited the υC=N stretching of amine functions (υC=N = 1593 cm^−1^ and 1585 cm^−1^, respectively (Figure 1)). Similar studies are in conformity with this [48,49,50]. Noticeably, the Si–O–Si spectrum in COF-S7 and COF-S12 was observed at the 1104 cm^−1^ and 1086 cm^−1^ absorption bands. Similar reported research by [51] is in agreement with this.

The NMR spectrum and ^1^H NMR spectra of ONPS and OAPS were examined in deuterium acetone (acetone-d_6_), Figure 4. In the ^1^H NMR spectrum of OAPS, the aromatic hydrogens of OAPS were observed to display a strong chemical shift (7.47–6.34 ppm) due to the electron-donor abilities of the substituents (NH_2_) on phenyl rings; those for the amino protons were in the range of 4.0–5.0 ppm, which is in a ratio of 2:1. A previous study by [52] is in agreement with this. Due to the electro-withdrawing effects of nitro groups (NO_2_) on the phenyl ring of the ONPS, the signals relating to aromatic hydrogen protons were found in the range of ONPS at lower chemical shift (8.7–7.8 ppm). Numerous aromatic signals were obtained in the ONPS band, notably triplet signals at (8.67 ppm) that were noticed to be protons between the nitro (NO_2_) and siloxy (Si–O–Si) groups in the meta isomer. Related research analyses were supported, in which the ONPS signals were characterized at a range of 8.7–7.8 ppm, while the range of OAPS was also discovered at a scale of 7.8–6.2 ppm and 7.8–6.0 ppm, respectively [51,54].

The polyhedron oligosilsesquioxane structural systems were preserved during the synthesis, as revealed by ^13^C NMR solid-state spectra for OAPS. At the end of the synthesis, signals from the trends assigned with ONPS were totally replaced by a new OAPS array peak position in the higher range of 118.0–152.9 ppm, indicating that the procedure was achieved (Figure 5). Four distinct aromatic carbon environment peaks were observed at 118.05, 132.39, 147.27 and 152.93 ppm, respectively. This demonstrates that the cage architecture of polyhedral oligosilsesquioxane remains stable during processing. Research by [53,54,55] displayed a similar trend.

The integrity of the oligomeric silesquioxane enclosure structure in the synthesis was also verified using a ^29^Si solid-state NMR spectrum for OAPS. The acquired results clearly demonstrated that the cage architecture of the polyhedral silesquioxane is preserved during synthesis, as shown in Figure 3, with OAPS signal locations of −79 and −69.6 ppm, respectively. This is consistent with recorded scientific investigations conducted using a traditional technique and a Pd/C catalyst [51,54,56].

^13^C NMR spectra of COF-S7 and COF-S12 were used to confirm the presence of various carbon environments in the synthesized materials (Figure 6). Aromatic carbons (phenyl rings) for the OAPS cluster and other linkers present in the synthesis of the nanomaterials were attributed to the overlapping patterns in the spectra at chemical shifts ranging from 112.6–139.3 ppm. The various resonating regions for COF-S7 and COF-S12 were investigated at the signals 126.02–145.3, and 117.5–139.3 ppm, respectively. Due to the impact of the aromatic linkers, the relevant chemical shifts were slightly altered to lower chemical shifts when the signals of OAPS and COFs were assessed. A related analysis performed by Hoffman and colleagues is in agreement with this [57]. Moreover, the chemical shifts of the azomethine carbons (C=N) at distinct vibration ranging from 161.9–165.3 ppm were disclosed by the synthesized COFs. In COF-S7 and COF-S12, the C=N signals were observed at 161.9 and 165.3 ppm, respectively. Consistent readings have been documented by [57,58]. Moreover, COF-S7 confirmed the signal at 23.6 ppm, indicating that the alpha aromatic methyl carbon group of the 2-methylanthraquionone linkage is responsible.

Furthermore, ^13^C CP MAS solid-state NMR showed a resonance of aromatic υC=N bond environments at 161.92 ppm and 165.39 ppm, respectively. Meanwhile, COF-S7 affirmed the signal at 23.6 ppm which was attributed to the alpha aromatic methyl carbon group (R–CH_3_) of the 2-methylanthraquionone linker. Moreover, the aromatic υC=C resonance was spotted for OAPS and TPA in the range 117.5–139.3 ppm. By comparing the signals the OAPS and COFs produced, the corresponding chemical shifts were slightly shifted to lower chemical shifts owing to the influence of the aromatic linkers.

The presence of silicon in cubic silesquioxane was revealed by solid-state ^29^Si MAS CP-NMR, which was used to evaluate the synthesis of the novel materials. As shown in Figure 6, the evidence gathered demonstrated the efficacy in the synthesis. For COF-S7 and COF-S12, magic-angle spinning provided a single signal with chemical shifts of −81.02 and −79.06 ppm, respectively. The Si^8^ vertices were found throughout the COF architectures as a result of these findings. The cubic silesquioxane vertices from the precursor material (OAPS) were all oriented in a certain direction, which allowed for the observation of a single chemical shift. Even after the synthesis of COFs, the analysis validated the cage architectures’ stiffness and regularity. This finding corresponds with other investigations on the synthesis of COF materials containing silicone [59,60].

TGA examinations of as-synthesized COF-S7 and COF-S12 compounds were assessed at temperatures ranging from 50 to 800 °C min^−1^ and a thermal rate of 10 °C min^−1^ under oxygen as in Figure 7. COF-S7 indicated a continuous weight loss in the ranges of 50–145 °C and 145–276 °C, which could be due to the elimination of water molecules and highly coordinated solvent, which were connected to weight losses of 14.83 and 21.35 wt%, respectively. As a result, full thermal decomposition is achieved at temperature of about 300–350 °C and at 54.8%. Interestingly, COF-S12 demonstrated an exceptional thermal stability up to 300–400 °C before collapse of the framework, consequential in a mass loss of 44.2 wt%. The product declined in weight from 50–150 °C, with a weight loss of 16.17 wt% assigned to the loss of water molecules, and further deteriorated from 150–260 °C, with a weight loss of 21.2 wt% related to strong solvent removal. The related literature is consistent with these findings [61,62,63,64], with a percentage loss of roughly 54.8 wt% when the structure disintegrates. (Figure 7).

The synthesized compounds obtained displayed the morphological components of the materials via FESEM images in which COF-S7 revealed itself to be even and highly sphere-shaped with a diameter of 100 nm and a rough surface micro-grain porosity crystal-size arrangement as shown in Figure 8. This is comparable with the results of [65], who produced a QD-doped COFs@MIP containing 1,3,5-triformylphloroglucinol (TP) and phenylenediamine (Pa). Similarly, COF-S12 revealed a nanocrystal porosity with a uniform homogeneous cubic morphology and diameters of 1µm and 100 nm, respectively. The same architecture size was observed in a study [60], in which a SiCOF-5 was produced by condensation by reticulating dianionic hexacoordinate [SiO6]^2−^ nodes.

Energy-dispersive X-ray (EDX) analysis confirmed the stoichiometric quantity as well as the morphological structure of the components present in the synthesized materials as shown in Figure 9. The elemental constituents (%) displayed atomic peaks that proved the presence of the elements (C, N, O and Si) in percentages, which further confirmed the synthesis of the new COFs. COF-S7 exhibited all of the predicted elements, with 55.98, 13.51, 22.27 and 5.25 in atomic % and 66.65, 7.06, 12.10 and 14.16 in weight % for C, N, O and Si, respectively. Similarly, COF-S12 indicated uniform atomic % and weight % values of 57.25, 12.31, 24.99, and 5.45 and 56.03, 7.68, 13.17 and 23.11, respectively.

The as-synthesized nanomaterials were used for the measurement of the isotherm at 77 K from 0 to 1 bar (1 bar = Po), which indicated that all the COFs showed a Type IV isotherm, which is characteristic of mesoporous materials with pore sizes of lower than 50 nm. The hysteresis loops are known as capillary condensation, which occurs in mesopores with a regulated uptake of high P/P_O_ lower than 1. Furthermore, the nanoparticles from the adsorption isotherm indicated Type H3 hysteresis loops (Figure 10) in the range of 0.5–0.9 relative pressure P/P_O_ which implies an extensive dispersal in pore size [66]. A Type IV with a hysteresis loop was observed in a comparable study, which was designated by a rapid absorption under low relative pressures at P/P_O_ 0.01 followed by a second step in 0.05 P/P_O_ 0.20, which is suggestive of a mesoporous structure [67].

The COF-S7 and COF-S12 displayed a BET surface area of 3.44 and 20.73 m^2^ g^−1^, respectively (Table 1). Considering this low surface area, adsorption is likely owing to the surface tension that occurs during the activation of the COFs, as the low boiling point solvent utilized is likely to retain robust hydrogen bonding to the vertices of COFs. These diluents will produce sufficient capillary force, resulting in partial or complete disruption of COF scaffolds, or very poor N_2_ absorption and negligible surface area, which could be due to pore network obstruction by huge pieces [67,68]. This could be due to the strong polarity and hydrogen capacity of the alcohol group (OH), which makes COF activation challenging.

The removal efficiency and adsorption capacity of naproxen (NAP) (Figure 11) for the synthesized COF-S7 and COF-S12, respectively, showed a significant rise as early as 25 min in. After about 3 h, COF-S7 showed a consistent and gradual improvement in absorption, with naproxen exclusion reaching about 50% after 2 h. COF-S12, on the other hand, had a remarkable removal performance at 25 min, achieving over 50% reduction. This finding was found to be consistent with previous studies [45,69] in which a MOF was produced with a graphene oxide composite (GnO/MIL-101) for naproxen and ketoprofen adsorption elimination. Nevertheless, it was observed that COF-S12 absorbed considerable quantities of NAP, when no specific adsorption mechanism, such as van der Waals force, is available, and surface area is widely considered as one of the most basic features in adsorption [70]. Moreover, COF-S7 and COF-S12 adsorption capacities (qe) showed a maximum experimental adsorption ability (qmax) of 35 and 42, and a computed ability (qmax) of 11.71 and 24.60, respectively. Table 2 summarizes the maximum adsorption capability (qmax) and calculated capability (qmax) of the adsorbents.

The pH effects on the adsorption of naproxen on COF-S7 and COF-S12 were investigated within a pH range of 4–12. As illustrated in Figure 11, the qt (after 210 min) of COF-S7 for naproxen gradually declined with an upsurge in pH under the conditions investigated. However, COF-S12 exhibited a slow decline with an upsurge in pH, with a rate equilibrium time (qt) of 270 min. With the exception of COF-S12, the findings obtained for the as-synthesized COF-S7 showed that there is steady exclusion as the pH rises to 9. The drop value was attained at pH 9 due to the additional negative charge caused by repulsion between the generated COFs and the naproxen solution.

The electrostatic interface, or π–π interactions, which are commonly used to depict the adsorption of numerous organic molecules such as phthalic acid, 2,4-dichlorophenoxyacetic acid, and p-arsanilic acid, may be seen as a reliable rationale in known adsorption processes [71,72], due to the electrostatic repulsion of the anionic form of NAP and the deprotonated hydroxyl group (pKa = 4.2) incorporated in the nanomaterials [73]. The qt levels ought to fit less at a pH > 4.2 if electrostatic correlation is the prime mechanism. Despite the fact that the qt rates deteriorated with pH, NAP was adsorbed in extensive magnitudes at pH > 4.2. as formerly affirmed [73,74]. As a consequence, a mechanism other than electrostatic exchange is suitable to describe NAP adsorption.

Depending on the presence of many active groups, H-bonding could be a viable adsorption mechanism for liquid-phase adsorption, particularly for water treatment [69,75,76]. The qt variables were employed to control the path of H-bonding between NAP and the as-synthesized COFs throughout a wide pH range. NAP dwells in a deprotonated state in the pH scale studied (pKa = 4.2) due to its pKa value. As a consequence, NAP should be used as the H-bond acceptor, and COFs produced as the H-bond donor. Interestingly, owing to further deprotonation of the hydroxyl group (pKa 10.0), the qt in COF-S7 nanoparticle decreased significantly at pH 10.0 [74]. It is worth noticing that the number of hydroxyl groups in COF-S12 was expressively better than in other bonding locations. COF-S12 adsorbed less NAP than COF-S7 at pH 11 or higher values. This is probably due to the full deprotonation of the hydroxy assembly in COF-S12 to produce –O^−^ at pH 10, resulting in repulsive interaction between –O^−^ and the COO^−^ group of NAP. In effect, COF-S12 displayed a lesser NAP qt than COF-S7 (at pH 11). Consequently, H-bonding (H-bond donor: COF-S12, H-bond acceptor: NAP) can illuminate the experiential adsorption of NAP throughout an extensive pH range. A related drift of naproxen removal efficacy at higher pH values was established in preceding findings [45,69].

The dose correlation coefficient R^2^ values of COF-S7 and COF-S12, as shown in Figure 12a,b, has verified the Freundlich adsorption isotherm as being favorable in determining the efficacy of the as-synthesized materials for the removal of naproxen. The adsorption level of connection among adsorption and solution concentration is also dependent on the adsorption intensity, n. As a result, these materials were shown to be a chemically favorable approach, with adsorption strengths less than 1 (n < 1). Table 2 shows the values of n, 1/n, K_F_, K_L_ and R^2^ for the current analysis.

Reusability is one of the most important criteria in the commercialization of adsorbents since it is directly related to the adsorption procedure’s outlay efficacy. Therefore, the regeneration of COF-S7 and COF-S12 was investigated after washing the used adsorbent with ethanol. Desorption is anticipated to restore the new adsorbent’s ability. The regeneration evaluation was observed in five adsorption/desorption cycles [46,77] as demonstrated in Figure 13. With a rise in the number of recycles, COF-S7 recorded the least loss in adsorption cycles, at 71.2 percent. In comparison, COF-S12 exhibited an enhanced performance (82.8%), with no significant decline in adsorption efficacy as the number of recycles increased, indicating that the material may be regenerated multiple times without significant adsorption degradation. This result was in agreement with the research recorded by [69]. Reflecting their facile synthesis, significant adsorption efficiency, and commendable regeneration, the as-synthesized materials are anticipated to be a prospect adsorbent for eliminating AIDs from water.

## 4. Conclusions

The COFs synthesized demonstrated eight connected octahedral nets from the eight nitrogen atoms from the OAPS assembly. Interestingly, the materials synthesized displayed a new topology with a 2D underlayer arrangement that was not predictable before using the Reticular Chemistry Structure Resource (RCSR). A UV/spectroscopy system was used to study the adsorption of naproxen on various adsorbents. A significant number of adsorbates were eliminated after 270 min. Likewise, the degree of correlation (R^2^) level suggested that the adsorption paradigm was compatible with the Freundlich isotherm model (with a higher value of R^2^ in the Freundlich model as compared to the Langmuir model). This also means that for the materials synthesized, the model of adsorption intensities (n) favored the chemical process, as we have confirmed (n < 1). As a result of this extensive assembling and adsorption recyclability study, POSS COFs may be employed with impressive results in the future for the adsorption treatment of pharmaceutical waste.

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
