# Peer review of "Adsorptive Removal of Naproxen from Water Using Polyhedral Oligomeric Silesquioxane (POSS) Covalent Organic Frameworks (COFs)"

_nanomaterials, 2022, doi:10.3390/nano12142491_

Round 1

Reviewer 1 Report

Nanomaterials-1712047 reports a polyhedral oligomeric silesquioxane (POSS) covalent organic frameworks (COFs) for adsorptive removal of naproxen, which is an anti-inflammatory drug. The adsorbent is novel, and the data is abundant. I recommend to accept this manuscript after addressing some important points.

Abstract, page 1. The logical relationship of the Abstract is weak, since there is no correlation between the COFs materials and naproxen. The authors should we consider using such adsorbent for removal of naproxen.

Introduction.

Page 2, line 61-63. The concept of photodegradation, photo transformation, and advanced oxidation processes are overlapped. In my head, advanced oxidation processes include some photocatalytic process, so it is better to re-organize these sentences.

Page 2, line 66-69. The classification of adsorbents is a bit weird. What does nanoporous substances mean? Please add more discussion on the comparison between MOFs and COFs.

Scheme 1. The X-axis of the adsorption isotherm is Ce (the concentration of naproxen at equilibrium), not time!

Page 4, line 140. Rain water? Why should we mimic the pH of rain water?

Page 5-6, line 186-190. The results of chemical stability of materials should be analyzed in detail. Blunt question. What’s the optimum solvent for activation of COF-S7 and COF-S12, respectively?

Page 11, line 332-335. The BET surface area of COF-S7 and COF-S12 are low. It’s interesting. The reviewer thought the surface area of COFs are high because of their porous structure.

Figure 12. The data doesn’t fit well with the Langmuir and Freundlich models. Please try other models.

Have you tested the effects of some co-existing ions and humic acid on the removal of naproxen by COF-S7 and COF-S12?

Author Response

Reviewer’s Comments (Reviewer 2)

Response

1)      Pharmaceuticals and personal care products (PPCPs) have emerged as a new threatened species.” is wrong, PPCPS threaten the environment, not the other way round. Furthermore, naproxen does not belong to the group of emerging toxins, those are in fact – toxins (cyanogen glucosides, alkaloids, etc.) 

2) What is meant by “strong reversible bonds”? Please elaborate.

3) Naproxen cannot be generated in the environment, because it is anthropogenic

4) Substantial execution was noted in the amputation ability of the naproxen” – this is incomprehensible

5)The synthesis part in the abstract is convoluted. It is written as if the precursors were synthesized as well.

6) “permanence” (page 2, line 58), please use a more conventional term “persistence”

7) naproxen and ketoprofen should be written with lowercase letters

8) Scheme 1 does not show an isotherm, but kinetics of naproxen removal.

9) what is a “naked round-bottom container”, page 3, line 105?

10) “generated about 40%”, page 3, line 115, means “yield” or?

11) no procedures related to characterization (FTIR, FESEM, XRD, TGA, NMR) were described in the Materials and methods.

12) what is meant by “hump-ordered”, page 5 line 175? The diffractogram of COF-S12 displays a notable amorphous halo. I believe that is that “hump” the authors refer to. Please comment.

13) how were the diffraction spectra simulated? Amend the materials and methods.

14) on Scheme 2 COF-S4 is displayed, however in this paper only COF-S7 is mentioned. I believe this is an error. Please amend.

15) the font size of tick mark labels and axis labels on Figs. 3, 4 and 5 need to be larger

16) how structural integrity can be evaluated by FTIR and other methods? “Structural integrity” implies mechanical properties, not chemical properties! Amend.

17) the termogram on Fig. 7(a) shows a weight loss for COF-S7 of nearly 100%. How is that possible, if COF-S7 should yield a SiO2 due to silesquioxane?

18) what is meant by the sentence on page 8, lines 245-246? I don’t see an obvious link between Pd/C and investigated COFs.

19) why is the stability of COFs considered exceptional up to 600 ℃, if they recorded mass loss was greater than 80% up to that temperature for both COFs?

20) EDX mapping should be provided alongside on Figure 9.

21) the stability of COFs in dilute aqueous acids and bases should have been investigated and provided, as this is an important practical aspect for their potential usage.

22) RCSR is mentioned for the first time in the conclusions. It should be discussed previously.

23) The cited literature references should be newer, most of the cited papers are older than 5 years.

The sentence has been re-shuffled.

The term (strong reversible bonds) has been elaborated.

Naproxen can be generated in the environment.

Similar execution of Naproxen amputation ability was documented by (M. Sarker et al. 2018; B. Feng et al., 2018; Zhuang et al., 2020).  

Yes, the precursors ONPS, OAPS were synthesized from the OPS which was originally purchased.

The term “permanence” was replaced

The replacement was done.

Adsorption isotherm was replaced by the removal kinetics

It’s a “3 naked round-bottom container”

It was corrected.

Yes, I meant yield.

The procedures were included

The hump-ordered means a reflecting lattice plane from the scattering pattern produced when a beam of radiation or particles interacts with it.

No, to check the nature of the material, the Bragg ’peaks appearing in the XRD pattern of amorphous with short range ordering, you get a broad hampered peak and if you have a sharp peak, then the material is crystalline.

The, method for the simulation is included

It’s an error, it was corrected.

The figure labels and axis were adjusted

It has been amended.

The overall weight loss of COF-S7 was at about 54.8 %.

Pd/C is a catalyst used for hydrogenation

No, it’s was corrected between 300 – 350 ℃ and 300 -400 ℃ for COF-S7 and COF-S12 respectively.

It’s done.

It’s important. However, this research focused on protic and aprotic solvents.

That will probably be conducted in future work.

Reticular chemistry structure resource (RCSR) was formally discussed.

Some of the cited literatures were the first researchers to synthesized and published the COFs materials.

Reviewer 2 Report

Suleiman et al. have prepared covalent organic frameworks (COFs) for the use as adsorbents for the removal of naproxen and ketoprofen in water. The submitted manuscript has some merit, as I find the presented COFs to be interesting, however, it is so poorly written that the submitted paper presents a major conundrum for the reviewer, and potential reader alike. So much so that I recommend that the paper be rejected because the poor quality of written English greatly affects the comprehensibility of the paper altogether. I am perplexed because the authors seem to have avoided using common terminology and come up with their own instead. In addition, stability of COFs in dilute acidic and alkaline conditions should have been investigated, as that is an important property for the usage of COFs in the role of adsorbents in aqueous medium. The changes required are too extensive to make a recommendation for major revision, I am afraid. I suggest that the authors consult a more proficient English speaker to help them in formulating their ideas and resubmit the manuscript later.

In light of this, I have some specific remarks, which are by no means extensive, but may help the authors towards a right direction:

1) “Pharmaceuticals and personal care products (PPCPs) have emerged as a new threatened species.” is wrong, PPCPS threaten the environment, not the other way round. Furthermore, naproxen does not belong to the group of emerging toxins, those are in fact – toxins (cyanogen glucosides, alkaloids, etc.)  

2) What is meant by “strong reversible bonds”? Please elaborate.

3) Naproxen cannot be generated in the environment, because it is anthropogenic

4) “substantial execution was noted in the amputation ability of the naproxen” – this is incomprehendable.

5) the synthesis part in the abstract is convoluted. It is written as if the precursors were synthesized as well.

6) “permenance” (page 2, line 58), please use a more conventional term “persistence”

7) naproxen and ketoprofen should be written with lowercase letters

8) Scheme 1 does not show an isotherm, but kinetics of naproxen removal.

9) what is a “naked round-bottom container”, page 3, line 105?

10) “generated about 40%”, page 3, line 115, means “yield” or?

11) no procedures related to characterization (FTIR, FESEM, XRD, TGA, NMR) were described in the Materials and methods.

12) what is meant by “hump-ordered”, page 5 line 175? The diffractogram of COF-S21 displays a notable amorphous halo. I believe that is that “hump” the authors refer to. Please comment.

13) how were the diffraction spectra simulated? Amend the materials and methods.

14) on Scheme 2 COF-S4 is displayed, however in this paper only COF-S7 is mentioned. I believe this is an error. Please amend.

15) the font size of tick mark labels and axis labels on Figs. 3, 4 and 5 need to be larger

16) how structural integrity can be evaluated by FTIR and other methods? “Structural integrity” implies mechanical properties, not chemical properties! Amend.

17) the termogram on Fig. 7(a) shows a weight loss for COF-S7 of nearly 100%. How is that possible, if COF-S7 should yield a SiO2 due to silsesquioxanes?

18) what is meant by the sentence on page 8, lines 245-246? I don’t see an obvious link between Pd/C and investigated COFs.

19) why is the stability of COFs considered exceptional up to 600°C, if they recorded mass loss was greater than 80% up to that temperature for both COFs?

20) EDX mapping should be provided alongside on Figure 9.

21) the stability of COFs in dilute aqueous acids and bases should have been investigated and provided, as this is an important practical aspect for their potential usage.

22) RCSR is mentioned for the first time in the conclusions. It should be discussed previously.

23) The cited literature references should be newer, most of the cited papers are older than 5 years.

Round 2

Reviewer 2 Report

The authors have made changes to the manuscript and improved the original draft. However, there are still some open questions.

1) The authors may not have understood my initial comment regarding "amputation". "Amputation" means specifically the removal of a limb or body part. I do not see any body parts removed in this manuscript. The references in the response in favour of "amputation" should have at least included the journal name and volume, as it is difficult to find the exact papers the authors had in mind. A suitable replacement would be "removal".

2) The authors responded that the weight loss of COF-S7 during TGA analysis was ~54.8%. The y-axis is labelled "weight loss %" and the weight loss decreases with temperature. I know you can see this in some papers, but it is meaningless. The y-axis represents the change in weight at a given temperature relative to the initial weight at the initial temperature, i.e., the y-axis should simply be "weight %". Nevertheless, the curve approaches 0% at 800 °C, so I wonder where the silica from the OAPS precursor disappeared in the case of COF-S? EDX, NMR and FTIR prove that there is silica inside. On the other hand, COF-S12 has some residual mass at 800 °C. The author's answer is insufficient.

3) the quality of some of the figures needs to be improved as they are difficult to read on screen (Figs.: 6, 9, 11, 12).

Author Response

Please see the attachment uploaded as a PDF file. Thank you 
